# Measurement of subglottic diameter and distance to pre-epiglottic space among Chinese adults

**Wai-Ho Chan[1], Chih-Wei Sung[2], Herman Chih-Heng Chang[3], Patrick Chow-In Ko[4], Edward Pei-Chuan Huang[4], Wan-Ching Lien[4]\*, Chien-Hua Huang[4]**

**1** Department of Emergency Medicine, Tan Tock Seng Hospital, Singapore, Singapore, **2** Department of Emergency Medicine, National Taiwan University Hospital Hsin-Chu Branch, Hsinchu, Taiwan, **3** Department of Emergency Medicine, Jinshan branch, National Taiwan University Hospital, New Taipei City, Taiwan, **4** Department of Emergency Medicine, National Taiwan University Hospital and National Taiwan University, Taipei, Taiwan

\* wanchinglien@ntu.edu.tw

**Data Availability Statement:** All relevant data are within the manuscript and its Supporting Information files.

## Abstract

Proper endotracheal tube (ETT) size selection and identification of potentially difficult airways are important to reduce laryngeal injury during intubation. However, controversies exist concerning transverse subglottic diameter—the narrowest part of the airway—and the distance to pre-epiglottic space. Because few studies have reported the distance from skin to the midpoint of the epiglottis (DSE) among normal individuals, whether the DSE varies between individuals and by ethnicity remains uncertain. The present study aims to investigate the sonographic subglottic diameter and DSE among healthy Chinese adults. Healthy volunteers were recruited at National Taiwan University Hospital between October and November 2019. Exclusion criteria included pre-existing airway or respiratory diseases, neck tumors, and a history of neck operation. Age, sex, height, weight, body mass index (BMI), sonographic DSE, and transverse subglottic diameter were recorded. A total of 124 participants were enrolled. The average age was 32.5 ± 10.4 years and 63 participants (51%) were males. The subglottic diameter was positively associated with sex (males, 14.40 mm; females, 11.10 mm, $p < 0.001$) and BMI (underweight, 12.13 mm; normal weight, 12.47 mm; overweight, 13.80 mm; obese, 13.67 mm, $p = 0.007$). Moreover, the DSE was shorter in males (male, 16.18 mm; females, 14.54 mm, $p < 0.001$) and participants with increased BMI (underweight, 13.70 mm; normal weight, 15.06 mm; overweight, 16.58 mm; obese, 18.18 mm, $p < 0.001$). As compared with other ethnicity, a smaller size of subglottic diameter and a shorter DSE were noted among Chinese participants, and we suggest that a relatively smaller size of endotracheal tube selection should be considered in tracheal intubations.

## Introduction

Airway management is essential in emergency and critical care settings. In all airway emergencies, a definite or secured airway brings advantages for patients' ventilation over a bag-mask

**Funding:** The authors received no specific funding for this work.

**Competing interests:** The authors have declared that no competing interests exist.

device. An endotracheal tube (ETT) delivers a high concentration of oxygen, referring to a definite airway. However, difficulties have occurred in 10%–19% of intubations [1–3]. An unexpected failure of intubation would lead to repeated attempts and possibly result in serious complications, including airway trauma, autonomous nervous system-related cardiac arrhythmia, and hypoxemia-related cardiac arrest [4]. Moreover, it could negatively impact vocal fold mobility, ventilation, and even quality of life [5, 6].

There are two possible ways to minimize airway injury during intubation: proper ETT size selection and preliminary recognition of difficult airways [7]. ETT-associated laryngeal injury occurs mostly at the level of the cricoid cartilage, the smallest diameter of the normal upper airway [8]. However, current recommendations for ETT size selection are based on previous cadaveric studies [9, 10]. Evidence regarding patient-specific ETT size selection remains limited [6]. Traditional screening tests for difficult airways, such as the Mallampati score, interincisor distance, thyrohyoid distance, chin-to-hyoid distance, and body mass index (BMI), have been used; however, the sensitivity and specificity have varied [11, 12]. The modified Look-Evaluate-Mallampati-Obstruction-Neck mobility score has limitations in patients with severe and complex trauma, such as massive bleeding and poor visibility fields of the mouth, neck, or face [13]. The Cormack–Lehane classification is considered a useful method for prediction of difficult laryngoscopy, but it requires direct visualization of the upper airway, which may not be feasible during emergency intubations [14].

Ultrasonography (US) is a real-time, noninvasive, readily accessible diagnostic tool. It has a wide range of applications for airway management and can be used to identify anatomical structures in the upper airway [15, 16]. US can be used for assessment of the upper airway's narrowest diameter, the subglottic diameter, at the cricoid level to select the ETT size [17–19]. It has been shown to have a strong correlation with magnetic resonance imaging when measuring the subglottic diameter [16]. However, most of these studies have considered Western populations [16, 20–22]. A relatively smaller size of the subglottic dimensions in an Indian population was reported, possibly resulting in a higher incidence of laryngotracheal injuries [23].

Additionally, the distance from skin to the midpoint of the epiglottis (DSE) in certain patients can be adequately visualized using US [3, 15, 16, 24]. Previous studies have shown that DSE is a potential predictor of a Cormack–Lehane grade of at least 2b through direct laryngoscopy, and hence of difficult intubations [3, 14]. However, data on DSE among normal individuals remain limited in published reports, and whether DSE varies according to ethnicity remains uncertain. Therefore, we conducted a prospective, observational study to investigate the transverse subglottic diameter and the DSE among healthy Chinese adults.

## Materials and methods

### Study design and participants

This was a prospective, single-center, observational study, conducted at National Taiwan University Hospital (NTUH) between October and November 2019. It was approved by the Institutional Review Board of the Ethics Committee of NTUH (201910015RINC) and registered at ClinicalTrials.gov (NCT04175483). Written informed consent was obtained from the participants.

Adult healthy volunteers (aged older than 20 years) were recruited. Exclusion criteria included pre-existing airway or respiratory diseases, neck tumors, and a history of neck operation. The primary investigator was not involved in the recruitment process and had no prior knowledge of the recruitment method.

## Airway measurement

The airway dimension was measured by two independent physicians, who had been previously instructed on airway US in a 1-hour standard lecture and 8-hour practice. Both sonographers were supervised by a senior instructor who was certified by the Taiwan Society of Ultrasound in Medicine and had over 10 years of experience in sonographic examinations.

An SSA-780A ultrasound scanner (Canon, Japan) equipped with a 7–12 MHz linear transducer was used. The participants lay in a supine position with a slight neck extension. The thyroid cartilage and cricoid cartilage were identified by using two fingers. A linear probe was placed transversely on the cricoid cartilage (Fig 1A). The individual in this manuscript has given written informed consent (as outlined in PLOS consent form) to publish these case details. The mucosa–air interface, a hypoechoic edge, was recognized and the transverse subglottic diameter was measured (Fig 1B) because the transverse diameter is smaller than the anteroposterior diameter [16]. DSE was defined as the distance between the skin and the midpoint of the epiglottis (Fig 1C). Measurements by the two sonographers were recorded and averaged.

## Data collection

Age, sex, height, weight, BMI, subglottic diameter, and DSE were recorded. Any identifiable information was removed from the analysis by an independent investigator. Age was categorized into three groups: young (<35 years old), moderate (35–55 years old), and senior (>55 years old). BMI was divided into four groups based on the World Health Organization (WHO) classification of BMI [25]: underweight (≦18.5 kg/m$^2$), normal weight (18.5–24.9 kg/m$^2$), overweight (25.0–29.9 kg/m$^2$), and obese (≧30.0 kg/m$^2$).

## Statistical analysis

The sample size was estimated for the primary outcome by using PASS 2019 software (NCSS software, Kaysville, Utah, USA). The effect size of 0.37 from a previous study [23] was calculated and used for the subglottic diameter versus sex. The calculated sample size was 40 with a power of 0.9 and a 5% significance level. An effect size of 0.3 for Pearson correlation in cricoid versus height and weight (BMI) was used, and the calculated sample size was 112 with a power of 0.9 and a 5% significance level. For interrater reliability between the two sonographers, Cohen's kappa statistic () was calculated.

Categorical variables (presented as numbers and percentages) were compared between the groups by using a Chi-squared test. Continuous variables with normal distribution (presented as mean ± standard deviation) were compared between the BMI groups by using analysis of

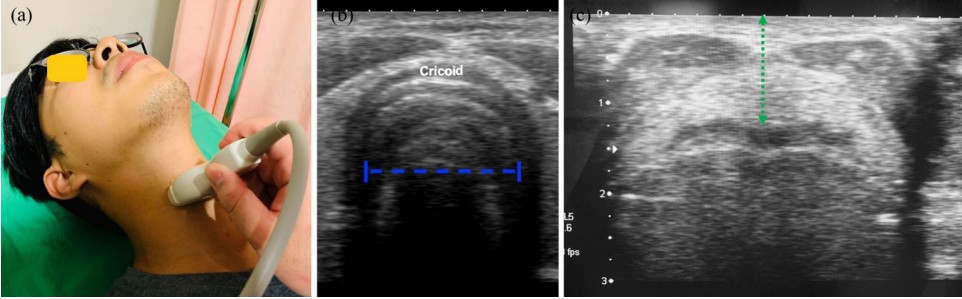

**Fig 1. Subglottic diameter measurement.** (a) The patient lies supine and the probe is transversely positioned at the subglottic region; (b) the measurement of the subglottic diameter (blue dashed line); (c) the DSE (green dotted line).

variance. Tukey's honestly significant difference post hoc test was subsequently performed for homogeneity of variances. A correlation analysis was performed and the Pearson's correlation coefficients (r) were determined between sex, BMI, and variables of airway dimension. A linear regression analysis was performed, and the best-fit regression model was demonstrated. A p-value of less than 0.05 was considered statistically significant. All statistical analyses were performed using SAS (version 9.4, Chicago, IL).

## Results

All of the 124 recruited volunteers were of Chinese ethnicity. Table 1 presents a comparison of the baseline characteristics of the participants. Of them, 63 participants (50.8%) were males. The mean age was 32.5 ± 10.4 years, with ages ranging from 19 to 74 years. The mean BMI was 23.1 kg/m$^2$, and BMI ranged from 15.0 to 35.4 kg/m$^2$. The mean subglottic diameter was 12.8 ± 2.0 mm. The mean DSE was 15.4 ± 2.1 mm, and DSE ranged from 11.2 to 21.0 mm. The -value between the two sonographers reached 0.87, indicating excellent interrater reliability.

The subglottic diameters of males and females are compared and illustrated in Fig 2. The male participants had larger subglottic diameter than the females (14.44 mm *vs.* 11.08 mm, $p < 0.001$) (Fig 2A). In total, 16 (12.9%), 74 (59.7%), 28 (22.6%), and 6 (4.8%) participants were underweight, normal weight, overweight, and obese, respectively. A higher BMI was significantly associated with a greater subglottic diameter (underweight: 12.13 mm; normal weight: 12.47 mm; overweight: 13.80 mm; and obese: 13.67 mm; $p = 0.007$) (Fig 2B). A total of 89 participants (71.8%) were younger than 35 years, whereas 27 participants (21.8%) were of moderate age (35–55 years old), and the remaining eight participants (6.4%) were seniors. However, no significant difference was observed between different age groups in terms of subglottic diameter ($p = 0.436$) (Fig 2C). Correlation analysis demonstrated that BMI was moderately and positively correlated with the subglottic transverse diameter ($r = 0.37$, $p < 0.001$). A linear regression model was constructed for estimating the subglottic diameter based on the following equation:

$$\text{Subglottic diameter (mm)} = 10.1 + 2.9 \times (\text{sex, male} = 1) + 0.04 \times \text{BMI}$$

Additionally, DSE was longer in males than in females and differed between different body sizes (Table 2). Patients with higher BMI had a longer DSE. A linear regression model was also constructed for estimating DSE based on the following equation (after adjusting for potential confounding factors):

$$\text{DSE (mm)} = 9.3 + 1.05 \times (\text{sex, male} = 1) + 0.25 \times \text{BMI}$$

**Table 1. Baseline characteristics and comparison of the DSE and subglottic diameter between sexes.**

| Variable | Total | Males | Females | *p* |
|---|---|---|---|---|
| | (N = 124) | (N = 63) | (N = 61) | |
| Age, years | 32.5 ± 10.4 | 32.0 ± 11.8 | 32.9 ± 9.3 | 0.485 |
| Height, cm | 166.7 ± 8.8 | 172.4 ± 6.3 | 160.8 ± 6.7 | < 0.001 |
| Weight, kg | 64.6 ± 14.6 | 72.2 ± 14.0 | 56.7 ± 10.4 | < 0.001 |
| Body mass index, kg/m$^2$ | 23.1 ± 4.0 | 24.2 ± 4.1 | 21.9 ± 3.4 | < 0.001 |
| DSE, mm | 15.41 ± 2.11 | 16.18 ± 2.03 | 14.54 ± 1.73 | < 0.001 |
| Subglottic diameter, mm | 12.80 ± 2.04 | 14.40 ± 1.13 | 11.10 ± 1.09 | < 0.001 |

Data are presented as mean ± standard deviation.

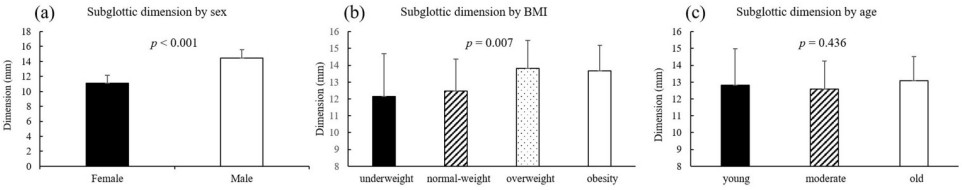

**Fig 2.** Comparison of subglottic diameter stratified by (a) sex, (b) BMI, and (c) age.

Table 3 presents a review of sonographic DSE and subglottic diameter from studies published between 1957 and 2019. Ethnicity, sex, age, and BMI were also listed if present in these studies. Males had a larger subglottic diameter than females in the Caucasian sample, as well as in our participants. Western populations had a longer mean DSE than that measured in our study.

## Discussion

This study demonstrated the feasibility of US for assessment of the subglottic diameter and DSE among healthy Chinese adults. Male sex and increased BMI were positively associated with a larger subglottic diameter and a longer DSE. Compared with previous reports among Caucasian samples [20, 21, 23], a relatively smaller subglottic diameter and a shorter DSE were noted among people of Chinese ethnicity in Taiwan.

The "size" of an ETT refers to its internal diameter. ETTs with sizes of 7.5 and 8.5 are recommended for adult female and adult male patients, respectively [26]. However, ETTs of sizes 7.0, 7.5, 8.0, and 8.5 have outer diameters (ODs) of 9.6, 10.2, 10.8, and 11.6 mm, respectively [18]. A previous study including Caucasians demonstrated adult subglottic diameters of 21.0 mm and 19.0 mm in males and females, respectively [20]. These diameters were larger than the ODs of the recommended ETTs [27]. By contrast, the OD of an ETT with a size of 7.5 is barely 1 mm smaller than the average subglottic diameter among normal Chinese females in this study. Oversized ETTs may cause complications such as pressure necrosis, tracheal stenosis, and obstruction [28–30], easily resulting in laryngeal injury when intubating with the current recommendations for ETT size. Furthermore, there is a greater risk of laryngeal edema or inflammation in patients requiring intubation, meaning that real patients would have higher chances of suffering from airway injury. On the other hand, if ETT size is too small, it can result in higher resistance of gas flow and more labored breathing, leading to intolerance to ventilator weaning [31, 32]. Therefore, it is important to choose a properly sized ETT for patients before intubation. The results in this study provide the normal airway size using the transverse subglottic diameter, and they imply that ETTs for Chinese adults should be a smaller size than that the current recommendations, such as a size of 6.5 to 7 for females and 7.5 to 8 for males.

**Table 2. Comparison of DSE between different body mass indices.**

|  | Body mass index | | | | |
|---|---|---|---|---|---|
|  | **Underweight** | **Normal-weight** | **Overweight** | **Obesity** |  |
| **Variable** | **N = 16 (12.9%)** | **N = 74 (59.7%)** | **N = 28 (22.6%)** | **N = 6 (4.8%)** | ***p*** |
| DSE, mm | 13.70 ± 1.45 | 15.06 ± 1.70 | 16.58 ± 2.08 | 18.18 ± 2.00 | < 0.001 |

Data are presented as mean ± standard deviation.

**Table 3. Literature review of DSE and subglottic diameter using ultrasonography.**

| Authors | Year | Population/Status | Male (n, %) | Age | BMI (male/female) | DSE (male/female) | Sub_D (male/female) |
|---------|------|-------------------|-------------|-----|-------------------|-------------------|---------------------|
| Jesseph et al. [20] | 1957 | Caucasian/NA | 21(45%) | 13–86 | NA | NA | 21.0/19.0 |
| Adhikari et al. [34] | 2011 | American/patients | 19 (37%) | 40–66 | NA | 23.7 | NA |
| Pinto et al. [35] | 2016 | Portuguese/patients | 39 (52%) | 37–73 | NA | 23.3 | NA |
| This study | 2019 | Chinese/volunteers | 63(51%) | 19–74 | 24.2/21.9 | 16.2/14.5 | 14.4/11.1 |

Measuring the subglottic diameter by using US provides a rapid, reliable method in emergency and critical care settings. The use of US to assess the airway and determine the size of the ETT is suggested before intubation because it can provide a rapid measurement of the airway, thus allowing better preparation of airway equipment and hence avoiding possible complications of over- or under-sizing of ETTs. Also, several US studies have been conducted regarding DSE and difficult intubations. Rana et al. reported that the ratio between DSE and distance from the epiglottis to the midpoint of the vocal cord was a good predictor of difficult laryngoscopy in 120 patients receiving elective surgery [33]. Gupta et al. reported a strong association between the pre-epiglottic distance and the Cormack–Lehane classification [14]. Falcetta et al. prospectively recruited 301 patients who underwent elective surgery, and observed that a threshold value of 2.54 cm for DSE was the best predictor of difficult intubations [3]. In the current study, the maximal dimension of DSE among healthy Chinese adults was 2.1 cm. Hence, a lower risk of difficult laryngoscopies or intubations may be speculated, although no data were available. Also, differences based on ethnicity may exist, because the DSE of Americans was longer than that of Asians (Indian and Chinese participants) [34–39].

Notably, both subglottic diameter and DSE were associated with sex and BMI in this study. The subglottic diameter was smaller in females than in males [17, 22, 40]. However, no related data for DSE or the effect of BMI were available. The present study provides evidence regarding independent predictors for subglottic diameter and DSE among adults of Chinese ethnicity. However, our findings require validation in future prospective multicenter and international studies.

Despite these contributions, there were some limitations in this study. First, this is a single-center observational study with convenience sampling instead of random sampling. However, the study exhibited adequate power and sufficient case numbers. Second, this study involved healthy volunteers. Difficulties could occur in performing the US airway evaluation under emergency conditions; for example, the airway might be obstructed by blood, or there could be a laryngeal fracture that would complicate measurement. Third, US assessment of the airway is operator-dependent and the measurement may vary between different operators with different levels of experience and training. However, after training and supervision by an instructor, sonographers can achieve good interrater reliability, as in this study. Fourth, US is a noninvasive method for airway measurement. Clinicians would use this method to first measure the subglottic diameter to help to select appropriate ETT size, followed by assessing the DSE to predict the possibility of difficult intubations. Finally, in each measurement, the pressure that the operator exerts on the anterior neck soft tissues may differ between patients. This interpatient and interoperator variation may cause random biases in the measurement of DSE.

## Conclusion

The study demonstrated the feasibility of US for assessment of the subglottic diameter for proper ETT size selection and assessment of the DSE for prediction of difficult intubations among healthy adults of Chinese ethnicity. These two parameters are positively associated with

male sex and higher BMI. Ethnic Chinese individuals exhibit a smaller subglottic diameter and a shorter DSE compared with existing evidence, so a smaller size of ETT is suggested among the Chinese population. Ethnic differences in risk of difficult intubation may exist. However, large-scale multicenter studies are required to validate the performance and potential impact of our findings on real patients. The predictive power of DSE combined with other parameters such as the modified Mallampati score should be investigated further.

## Supporting information

**S1 Data.**
(XLSX)

## Acknowledgments

The authors would like to acknowledge Dr. Tay Seow Yian and Dr. Ang Hou for their coordination of fellowship training.

## Author Contributions

**Conceptualization:** Wai-Ho Chan, Wan-Ching Lien.

**Data curation:** Wai-Ho Chan, Chih-Wei Sung.

**Formal analysis:** Chih-Wei Sung, Herman Chih-Heng Chang.

**Methodology:** Patrick Chow-In Ko, Edward Pei-Chuan Huang.

**Supervision:** Wan-Ching Lien, Chien-Hua Huang.

**Writing – original draft:** Wai-Ho Chan, Chih-Wei Sung.

**Writing – review & editing:** Wan-Ching Lien.

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
