## [Decision Letter · Decision Letter 0]

22 May 2020

PONE-D-20-07172

Measurement of subglottic diameter and pre-epiglottic space thickness Among Chinese adults

PLOS ONE

Dear Dr. Lien,

Thank you for submitting your manuscript to PLOS ONE. After careful consideration, we feel that it has merit but does not fully meet PLOS ONE’s publication criteria as it currently stands. Therefore, we invite you to submit a revised version of the manuscript that addresses the points raised during the review process.

We look forward to receiving your revised manuscript.

Kind regards,

Jorge Spratley, MD, PhD

Academic Editor

PLOS ONE

Journal Requirements:

a) Did participants provide their written or verbal informed consent to participate in this study?

b) If consent was verbal, please explain i) why written consent was not obtained, ii) how you documented participant consent, and iii) whether the ethics committees/IRB approved this consent procedure."

3. Thank you for including your ethics statement: 

"It was approved by the Institutional Review Board of the hospital (201910015RINC) and registered at the ClinicalTrials.gov (NCT04175483). Informed consents were obtained

from the participants.

Once you have amended this statement in the Methods section of the manuscript, please add the same text to the “Ethics Statement” field of the submission form (via “Edit Submission”).

4. We suggest you thoroughly copyedit your manuscript for language usage, spelling, and grammar. If you do not know anyone who can help you do this, you may wish to consider employing a professional scientific editing service.  

6. Thank you for stating the following financial disclosure:

"The funders had no role in study design, data collection and analysis, decision to

publish, or preparation of the manuscript."

7. PLOS requires an ORCID iD for the corresponding author in Editorial Manager on papers submitted after December 6th, 2016. Please ensure that you have an ORCID iD and that it is validated in Editorial Manager. To do this, go to ‘Update my Information’ (in the upper left-hand corner of the main menu), and click on the Fetch/Validate link next to the ORCID field. This will take you to the ORCID site and allow you to create a new iD or authenticate a pre-existing iD in Editorial Manager. Please see the following video for instructions on linking an ORCID iD to your Editorial Manager account: https://www.youtube.com/watch?v=_xcclfuvtxQ

8. We note that Figure 1 includes an image of a participant in the study. 

Reviewers' comments:

Reviewer's Responses to Questions

**Comments to the Author**

1. Is the manuscript technically sound, and do the data support the conclusions?

Reviewer #1: Partly

Reviewer #2: Yes

2. Has the statistical analysis been performed appropriately and rigorously? 

Reviewer #1: Yes

Reviewer #2: Yes

3. Have the authors made all data underlying the findings in their manuscript fully available?

Reviewer #1: Yes

Reviewer #2: Yes

4. Is the manuscript presented in an intelligible fashion and written in standard English?

Reviewer #1: No

Reviewer #2: Yes

5. Review Comments to the Author

Reviewer #1: This manuscript addresses the feasibility of ultrasonography in predicting difficult intubation among Chinese adults. A prospective, observational study was conducted to measure the ultrasonographic subglottic diameter and the “pre-epiglottic (PES) thickness” among healthy Chinese adults. The authors describe that the size of subglottic diameter and “PES thickness” is smaller compared to western populations.

In addition, they compare their results with previous reports and highlight that a smaller size of endotracheal tube should be selected in the Chinese population.

Unfortunately there are some fundamental concerns with the experimental design and, most critically, with the analysis and literature review.

1. The authors define pre-epiglottic thickness (PES) as the distance between the skin and the midpoint of the epiglottis. In fact, that is the measurement of the distance from the skin to the epiglottis as performed and written by other authors. The authors should speak in distance from skin to epiglottis (DSE) instead of PES. The title and manuscript should be revised accordingly to reflect the distance measured.

2. In addition, in ultrasound measurements of the anterior neck soft tissues the amount of pressure applied by the ultrasound probe can cause a difference in values and may alter the results. That fact is not stressed in the study limitations.

3. The authors revised literature between 1957 and 2019, not until 2016 as stated, and compared it with their results. Literature review and data comparison presented in, particularly in Table 3, has pitfalls:

a. The authours compare ultrasonography measurements with CT scan measurements [Kamel et al. (2009); Prasanna Kumar et al. (2014), Tai et al. (2016)].

b. In addition, Tai et al (2016) performed CT scans in Caucasian and Chinese patients, not only in Chinese patients as stated.

c. Pinto et al (2016) evaluated the distance from skin to epiglottis (DSE) in a sample of Portuguese population and not in American patients.

d. Hall et al (2018) measured accurately the pre-epiglottic space from the anterior surface of the epiglottis to the anterior surface of the strap muscles. The BMI presented in the table is the BMI range in Hall’s study, not the comparison between male and female BMI.

e. The study of Yadav et al (2019) measured the distance from skin to hyoid bone, skin to the thyrohyoid membrane and maximum tongue thickness, not the pre-epiglotic space.

4. The authors should clarify the comparison with previous reports among caucasion samples (L202-204): “… Comparing with previous reports among Caucasian samples [22, 23, 25], a relatively smaller subglottic diameter and a thinner PES were noted among Chinese …”

a. Kamel et al. (2009) studied the morphometry of human trachea by CT scans and did not include subglottic diameter measurements (Reference # 23)

b. Kumar Prasanna and Ravikumar (2014) studied an Indian population not Caucasian (Reference # 25).

5. References are not formatted according to PLOS reference style.

a. References must be carefully confirmed and reviewed.

b. The same reference is cited thrice and is presented with different formats:

i. 16. KE Y-T. Point-of-care ultrasound (POCUS) of the upper airway. Can J Anaesth. 2018 Apr;65:473-84.

ii. 21. You-Ten KE, Siddiqui N, Teoh WH, Kristensen MS. Point-of-care ultrasound (POCUS) of the upper airway. Can J Anaesth. 2018;65(4):473-84. Epub 2018/01/20. doi:10.1007/s12630-018-1064-8. PubMed PMID: 29349733

iii. 27. You-Ten KE, Siddiqui N, Teoh WH, Kristensen MS. Point-of-care ultrasound (POCUS) of the upper airway. Can J Anaesth. 2018;65:473-84.

Further specific points:

1. Consider revising L37-39. “The prospective study aims to investigate the sonographic subglottic diameter and the pre-epiglottic thickness among healthy Chinese adults”

2. Consider revising L59-60. “An endotracheal tube (ETT) or a supraglottic airway device delivers a high concentration of oxygen, referring as initial methods of managing the airway [1].”

3. The authors should also consider that the distance of the skin to the epiglottis combined with the modified Mallampati score, for example, improves the predictive power in a decision tree over either test alone.

4. In Figure 1 A: the probe is transversely positioned in a higher location than the subglottic region.

5. Consider revising Table 1: Height, Kg and BMI Kg/m2 to kg and kg/m2, respectively.

6. The authors should revise the language to improve readability.

Reviewer #2: This Manuscript represents a prospective study that analyzed the diameter of the subglottic and pre-epiglottic space in healthy Chinese adults in order to establish the average values for this ethnicity and compare it with standard measures of Caucasians. As the Authors have established average values among their patients’ population, they believe that this information could contribute to the proper choice of the size of endotracheal tube and predict difficult intubation in patients.

Overall, this study is interesting and presents current relevance. However, there are some issues that should be addressed prior to publication.

Abstract

1. The thickness of the pre-epiglottic space (PES) among normal individuals are still limited.

Please rephrase this sentence in order to explain more clearly the problematic regarding the thickness of PES.

2. A smaller size of subglottic diameter and PES thickness were noted among Chinese.

Please rephrase this sentence and indicate to what patients’ population you compared your results with.

Introduction

3. A definite airway could provide effective oxygenation and ventilation rather than a bag-mask device.

I would rephrase this sentence, indicating more clearly that in all airway emergencies, a secured airway brings advantages for patients’ ventilation over a bag-mask device

4. Moreover, it could negatively impact on glottis mobility, ventilation and even quality of life

Rephrase glottis mobility into a vocal fold motility

5. But controversies were reported by investigators in Hong Kong [26].

Please indicate these controversies and present the problematic.

Results

6. The PES in Western countries, such as the United States, was thicker than that in Asian countries (India and China).

Please present the data to support your statement.

Discussion

7. The “size” of an ETT refers to its internal diameter. ETTs with the size of 7.5 and 8.5 are recommended for female and male patients, respectively [30].

Please specify that these measurements refer to adult patients.

8. In addition, there is possibly a higher risk of laryngeal edema or inflammation in real patients when she/he needs intubation.

I suggest you to omit the use of the term ‘real’ when referring to the patients;

9. Please emphasize and integrate into the Discussion section the literature data that support the fact that the subglottic space diameter is a predictor of a correct ETT size

10. I would integrate the difficulties in performing an US airway evaluation in emergency as well, as there is rarely time, often the airway is obstructed by blood or there is laryngeal fracture, all factors that could make this measurement harder

11. ‘it would be validated in future multi-center and international studies’

I would add prospective to multi-center and international studies.

12. The manuscript would benefit from an English native speaker revision

13. It would be interesting to conduct in the future the same measurement preoperatively on patients with laryngeal or neck morbidity and to compare the measurements with current results, as well to correlate these measurements with the Cormack score.

14. Integrate, if data available, the incidence of subglottic /tracheal stenosis or intubation related injuries among Chinese

6. PLOS authors have the option to publish the peer review history of their article (what does this mean?). If published, this will include your full peer review and any attached files.

Reviewer #1: No

Reviewer #2: No

---

## [Author Response · Author response to Decision Letter 0]

30 Jun 2020

Jorge Spratley, MD, PhD

Academic Editor

PLOS ONE

 Re-submission of Revised Manuscript (PONE-D-20-07172R1) “Measurement of subglottic diameter and distance to pre-epiglottic space among Chinese adults” by Chen et al.

Dear Professor Spratley:

We would like to thank you and the reviewers to give us valuable comments. We revised the title as the reviewer’s suggestion. Also, we responded to the reviewers’ comments point-by-point as follows. This manuscript was again edited by Wallace Academic Editing for language usage, spelling, and grammar.

We again thank the reviewers for their efforts. A reply at your earliest convenience would be highly appreciated.

Sincerely yours,

Wan-Ching Lien, M.D., Ph.D.

Responses to the Journal Requirements

(We have marked our answers in blue color while keeping the original opinions in black)

Ans: We have revised the manuscript to meet the style requirements.

a) Did participants provide their written or verbal informed consent to participate in this study?

b) If consent was verbal, please explain i) why written consent was not obtained, ii) how you documented participant consent, and iii) whether the ethics committees/IRB approved this consent procedure."

Ans: The participants provided the written informed consents.

3. Thank you for including your ethics statement: 

"It was approved by the Institutional Review Board of the hospital (201910015RINC) and registered at the ClinicalTrials.gov (NCT04175483). Informed consents were obtained from the participants.

Ans: We have added the full name of the IRB (page 7). It was approved by the Institutional Review Board of the Ethics Committee of the National Taiwan University Hospital (201910015RINC) and registered at ClinicalTrials.gov (NCT04175483). Written informed consent was obtained from the participants.

4. We suggest you thoroughly copyedit your manuscript for language usage, spelling, and grammar. If you do not know anyone who can help you do this, you may wish to consider employing a professional scientific editing service. 

Ans: Thank you for your comments. A professional scientific editing service was done. We provided a marked copy and a clear copy.

5. We note that you have indicated that data from this study are available upon request. PLOS only allows data to be available upon request if there are legal or ethical restrictions on sharing data publicly. For information on unacceptable data access restrictions, please see http://journals.plos.org/plosone/s/data-availability#loc-unacceptable-data-access-restrictions. In your revised cover letter, please address the following prompts:

Ans: Thank you for your comments. We have uoloaded the relevant files.

6. Thank you for stating the following financial disclosure:

"The funders had no role in study design, data collection and analysis, decision to

publish, or preparation of the manuscript."

Ans: Thank you for your comments. We added the sentence in the manuscript (page 19) and also amended statements within cover letter. 

7. PLOS requires an ORCID iD for the corresponding author in Editorial Manager on papers submitted after December 6th, 2016. Please ensure that you have an ORCID iD and that it is validated in Editorial Manager. To do this, go to ‘Update my Information’ (in the upper left-hand corner of the main menu), and click on the Fetch/Validate link next to the ORCID field. This will take you to the ORCID site and allow you to create a new iD or authenticate a pre-existing iD in Editorial Manager. Please see the following video for instructions on linking an ORCID iD to your Editorial Manager account: https://www.youtube.com/watch?v=_xcclfuvtxQ

Ans: I have validated my ORCID in Editorial Manager.

8. We note that Figure 1 includes an image of a participant in the study. 

Ans: The written consent form was signed and uploaded. 

 

Response to the Reviewers

(We have marked our answers in blue color while keeping the original critique in black. Please also refer to our manuscript in which we put our changes either in a track copy or a clean copy.)

For reviewer #1

1. The authors define pre-epiglottic thickness (PES) as the distance between the skin and the midpoint of the epiglottis. In fact, that is the measurement of the distance from the skin to the epiglottis as performed and written by other authors. The authors should speak in distance from skin to epiglottis (DSE) instead of PES. The title and manuscript should be revised accordingly to reflect the distance measured.

Ans: Thank you for this valuable comment. We fully agree with this suggestion. The title of the manuscript was revised as “Measurement of subglottic diameter and distance to pre-epiglottic space among Chinese adults” and the term “PES” was replaced by DSE. All of the changes were presented in the revised manuscript. 

2. In addition, in ultrasound measurements of the anterior neck soft tissues the amount of pressure applied by the ultrasound probe can cause a difference in values and may alter the results. That fact is not stressed in the study limitations.

Ans: Thank you for this valuable comment. The pressure on the soft tissue in each patient could hardly be quantized and this would be a limitation. The following sentences would be found in the Discussion section (Page 17)

“Finally, in each measurement, the pressure that the operator exerts on the anterior neck soft tissues may differ between patients. This interpatient and interoperator variation may cause random biases in the measurement of DSE.”

3. The authors revised literature between 1957 and 2019, not until 2016 as stated, and compared it with their results. Literature review and data comparison presented in, particularly in Table 3, has pitfalls: 

a. The authours compare ultrasonography measurements with CT scan measurements [Kamel et al. (2009); Prasanna Kumar et al. (2014), Tai et al. (2016)].

b. In addition, Tai et al (2016) performed CT scans in Caucasian and Chinese patients, not only in Chinese patients as stated.

c. Pinto et al (2016) evaluated the distance from skin to epiglottis

(DSE) in a sample of Portuguese population and not in American patients.

d. Hall et al (2018) measured accurately the pre-epiglottic space from the anterior surface of the epiglottis to the anterior surface of the strap muscles. The BMI presented in the table is the BMI range in Hall’s study, not the comparison between male and female BMI.

e. The study of Yadav et al (2019) measured the distance from skin to hyoid bone, skin to the thyrohyoid membrane and maximum tongue thickness, not the pre-epiglotic space.

Ans: Thank you for your clear and valuable comments. We agreed with you that we revised the table and presented the studies using only ultrasonography.

4. The authors should clarify the comparison with previous reports among caucasion samples (L202-204): “…. Comparing with previous reports among Caucasian samples [22, 23, 25], a relatively smaller subglottic diameter and a thinner PES were noted among Chinese ….”; a. Kamel et al. (2009) studied the morphometry of human trachea by CT scans and did not include subglottic diameter measurements (Reference # 23) b. Kumar Prasanna and Ravikumar (2014) studied an Indian population not Caucasian (Reference # 25).

Ans: Thank you for your clear and valuable comments. We agreed with you that we revised the manuscript. 

5. References are not formatted according to PLOS reference style.

a. References must be carefully confirmed and reviewed.

b. The same reference is cited thrice and is presented with different formats:

i. 16. KE Y-T. Point-of-care ultrasound (POCUS) of the upper airway.

Can J Anaesth. 2018 Apr;65:473-84.

ii. 21. You-Ten KE, Siddiqui N, Teoh WH, Kristensen MS. Point-of-care ultrasound (POCUS) of the upper airway. Can J Anaesth.

2018;65(4):473-84. Epub 2018/01/20. doi:10.1007/s12630-018-1064-8.

PubMed PMID: 29349733

iii. 27. You-Ten KE, Siddiqui N, Teoh WH, Kristensen MS. Point-of-care ultrasound (POCUS) of the upper airway. Can J Anaesth. 2018;65:473-84.

Ans: Thank you for your clear and valuable comments. We agreed with you that we revised the reference.

Further specific points:

1. Consider revising L37-39. “The prospective study aims to investigate the sonographic subglottic diameter and the pre-epiglottic thickness among healthy Chinese adults”

Ans: Thanks for the suggestion. The original sentence in L37-39 have been modified to “The present study aims to investigate the sonographic subglottic diameter and DSE among healthy Chinese adults”, which could be found in Abstract (page 2).

2. Consider revising L59-60. “An endotracheal tube (ETT) or a supraglottic airway device delivers a high concentration of oxygen, referring as initial methods of managing the airway [1].”

Ans: To clarify the argument, we revised the sentence as your suggestion. The following sentence could be found in Introduction (page 4).

“An endotracheal tube (ETT) delivers a high concentration of oxygen, referring to a definite airway.”

3. The authors should also consider that the distance of the skin to the epiglottis combined with the modified Mallampati score, for example, improves the predictive power in a decision tree over either test alone.

Ans: We appreciated the suggestion. The distance of the skin to the epiglottis combined with the modified Mallampati score may increase the sensitivity as well as other diagnostic parameters. In this study, however, the modified Mallampati score was not recorded. Further study would be done to provide evidence of combinational use. We added the following sentence in Discussion section (page 18).

“The predictive power of DSE combined with other parameters such as the modified Mallampati score should be investigated further.”

4. In Figure 1 A: the probe is transversely positioned in a higher location than the subglottic region.

Ans: Thank you for careful review. We replaced the original panel by a new figure 1A. 

5. Consider revising Table 1: Height, Kg and BMI Kg/m2 to kg and kg/m2, respectively.

Ans: Thank you for the valuable comments. We have corrected the “Kg” and “Kg/m2” to “kg” and “kg/m2”, respectively. 

6. The authors should revise the language to improve readability.

Ans: Thank you for the valuable comments. This manuscript was again edited by Wallace Academic Editing to improve readability.

 

For reviewer #2

1. The thickness of the pre-epiglottic space (PES) among normal individuals are still limited. Please rephrase this sentence in order to explain more clearly the problematic regarding the thickness of PES.

Ans: Thank you for the valuable comments. To clarify the issue, we rephrased the sentence. The revised sentence was shown as follows and could be found in Abstract (page 2). 

“Because few studies have reported the distance from skin to the midpoint of the epiglottis (DSE) among normal individuals, whether the DSE varies between individuals and by ethnicity remains uncertain.”

2. A smaller size of subglottic diameter and PES thickness were noted among Chinese. Please rephrase this sentence and indicate to what patients’ population you compared your results with.

Ans: Thanks for the suggestion. We rephrased the original sentence by the following one that could be found in Abstract (page 3).

“As compared with other ethnicity, a smaller size of subglottic diameter and a shorter DSE were noted among Chinese participants,…”

3. A definite airway could provide effective oxygenation and ventilation rather than a bag-mask device. I would rephrase this sentence, indicating more clearly that in all airway emergencies, a secured airway brings advantages for patients’ ventilation over a bag-mask device

Ans: We appreciated this recommendation. We revised the sentence as follows and could be found in the revised manuscript (page 4).

“In all airway emergencies, a definite or secured airway brings advantages for patients’ ventilation over a bag-mask device.”

4. Moreover, it could negatively impact on glottis mobility, ventilation and even quality of life. Rephrase glottis mobility into a vocal fold motility

Ans: Thanks for the suggestion. We rephrased glottis mobility into a vocal fold motility (page 4).

5. But controversies were reported by investigators in Hong Kong [26]. Please indicate these controversies and present the problematic.

Ans: Thank you for the suggestion. To avoid confusion that Tai’s study focused on tracheal diameter identified by the CT, we removed the reference and the original sentence.

6. The PES in Western countries, such as the United States, was thicker than that in Asian countries (India and China). Please present the data to support your statement.

Ans: We revised the Table 3 that listed the studies using ultrasonography. Also, PES was changed to the distance from skin to the midpoint of the epiglottis (DSE). The DSE in western countries was longer than that in Asians. For example, Adhikari and Pinto respectively reported their results, indicating that an average of DSE was about 23 mm. In our study, the DSE was 16.2 mm in males and 14.5mm in female (Table 3). 

7. The “size” of an ETT refers to its internal diameter. ETTs with the size of 7.5 and 8.5 are recommended for female and male patients, respectively [30]. Please specify that these measurements refer to adult patients.

Ans: Thank you for the comment. We added the word “adult” into the original sentence to specify the sentence (page 14). 

“ETTs with sizes of 7.5 and 8.5 are recommended for adult female and adult male patients, respectively”

8. In addition, there is possibly a higher risk of laryngeal edema or inflammation in real patients when she/he needs intubation. I suggest you to omit the use of the term “real” when referring to the patients;

Ans: Thanks for suggestion. We deleted the word “real”. 

9. Please emphasize and integrate into the Discussion section the literature data that support the fact that the subglottic space diameter is a predictor of a correct ETT size

Ans: Thanks for suggestion. We added the sentence to emphasize and support the fact that the subglottic space diameter is a predictor of a correct ETT size. The following sentence could be found in Discussion section (page 14)

“A previous study including Caucasians demonstrated adult subglottic diameters of 21.0 mm and 19.0 mm in males and females, respectively. These diameters were larger than the ODs of the recommended ETTs”

10. I would integrate the difficulties in performing an US airway evaluation in emergency as well, as there is rarely time, often the airway is obstructed by blood or there is laryngeal fracture, all factors that could make this measurement harder

Ans: Thank you for your valuable comments. We added the difficulties in the limitations. 

11. it would be validated in future multi-center and international studies”; I would add prospective to multi-center and international studies.

Ans: We fully agreed the suggestion. We added the word “prospective” into the original sentence (page 16). 

12. The manuscript would benefit from an English native speaker revision

Ans: Thanks for the suggestion. This manuscript was again edited by Wallace Academic Editing to improve readability.

13. It would be interesting to conduct in the future the same measurement preoperatively on patients with laryngeal or neck morbidity and to compare the measurements with current results, as well to correlate these measurements with the Cormack score.

Ans: We appreciated this viewpoint. Further study could be addressed in preoperative patients.

14. Integrate, if data available, the incidence of subglottic /tracheal stenosis or intubation related injuries among Chinese

Ans: Thank you for this recommendation. In this study, the measurement was performed in healthy volunteers. As best of our survey, on subglottic /tracheal stenosis or intubation complications among Chinese, neither the current study nor previous report was noted.

---

## [Editor Report · Decision Letter 1]

7 Jul 2020

Measurement of subglottic diameter and distance to pre-epiglottic space among Chinese adults

PONE-D-20-07172R1

Dear Dr. Lien,

We’re pleased to inform you that your manuscript has been judged scientifically suitable for publication and will be formally accepted for publication once it meets all outstanding technical requirements.

Kind regards,

Jorge Spratley, MD, PhD

Academic Editor

PLOS ONE

Additional Editor Comments (optional):

Thank you for your careful revision of paper PONE-D-20-07172R1. The manuscript has much improved in readability and consistency.

One of the weaknesses of the study is that US in the neck, as in many other body regions, is strongly dependent from examiner to examiner. Therefore, harmonization of results is often times difficult to perform.

Despite this, your article has been accepted for publication at PlosOne.

Good luck!
---

## [Editor Report · Acceptance letter]

10 Jul 2020

PONE-D-20-07172R1 

Measurement of subglottic diameter and distance to pre-epiglottic space among Chinese adults 

Dear Dr. Lien:

I'm pleased to inform you that your manuscript has been deemed suitable for publication in PLOS ONE. Congratulations! Your manuscript is now with our production department. 

Kind regards, 

on behalf of

Professor Jorge Spratley 

Academic Editor

PLOS ONE